# Influence of a Functional Nutrients-Enriched Infant Formula on Language Development in Healthy Children at Four Years Old

**DOI:** 10.3390/nu12020535

**Published:** 2020-02-19

**Authors:** Ana Nieto-Ruiz, Estefanía Diéguez, Natalia Sepúlveda-Valbuena, Elvira Catena, Jesús Jiménez, María Rodríguez-Palmero, Andrés Catena, M. Teresa Miranda, José Antonio García-Santos, Mercedes G. Bermúdez, Cristina Campoy

**Affiliations:** 1Department of Paediatrics, School of Medicine, University of Granada, Avda. Investigación 11, 18016 Granada, Spain; ananietoruiz@gmail.com (A.N.-R.); estefaniadieguezcastillo@gmail.com (E.D.); ecatenaverdejo@gmail.com (E.C.); joseantonio_gsantos@outlook.es (J.A.G.-S.); mgbermudez@ugr.es (M.G.B.); 2EURISTIKOS Excellence Centre for Paediatric Research, Biomedical Research Centre, University of Granada, 18016 Granada, Spain; sepulveda.natalia@hotmail.com; 3Instituto de Investigación Biosanitaria ibs. GRANADA, Health Sciences Technological Park, 18012 Granada, Spain; 4Mind, Brain and Behaviour Research Centre-CIMCYC, University of Granada, 18011 Granada, Spain; acatena@ugr.es; 5Nutrition and Biochemistry Department, Faculty of Sciences, Pontificia Universidad Javeriana, Bogotá 110231, Colombia; 6Ordesa Laboratories, S.L., 08820 Barcelona, Spain; Jesus.Jimenez@ordesa.es (J.J.); Maria.Rodriguez@ordesa.es (M.R.-P.); 7Department of Biostatistics, School of Medicine, University of Granada, 18016 Granada, Spain; tmiranda@ugr.es; 8Spanish Network of Biomedical Research in Epidemiology and Public Health (CIBERESP), Granada’s node, Institute of Health Carlos III, 28029 Madrid, Spain

**Keywords:** infant formula, functional nutrients, breastfeeding, language development

## Abstract

Nutrition during early life is essential for brain development and establishes the basis for cognitive and language skills development. It is well established that breastfeeding, compared to formula feeding, has been traditionally associated with increased neurodevelopmental scores up to early adulthood. We analyzed the long-term effects of a new infant formula enriched with bioactive compounds on healthy children’s language development at four years old. In a randomized double-blind COGNIS study, 122 children attended the follow-up call at four years. From them, 89 children were fed a standard infant formula (SF, *n* = 46) or an experimental infant formula enriched with functional nutrients (EF, *n* = 43) during their first 18 months of life. As a reference group, 33 exclusively breastfed (BF) were included. Language development was assessed using the Oral Language Task of Navarra-Revised (PLON-R). ANCOVA, chi-square test, and logistic regression models were performed. EF children seemed to show higher scores in use of language and oral spontaneous expression than SF children, and both SF and EF groups did not differ from the BF group. Moreover, it seems that SF children were more frequently categorized into “need to improve and delayed” in the use of language than EF children, and might more frequently present “need to improve and delayed” in the PLON-R total score than BF children. Finally, the results suggest that SF children presented a higher risk of suffering language development than BF children. Secondary analysis also showed a slight trend between low socioeconomic status and poorer language skills. The functional compound-enriched infant formula seems to be associated with beneficial long-term effects in the development of child’s language at four years old in a similar way to breastfed infants.

## 1. Introduction

The first years of life are critical for the development of language skills, including learning to understand and speak language. Neural networks for language acquisition are fully formed before birth. In fact, infants can perceive and react to sound at 24 weeks gestation and, at 35 weeks, they begin to learn language in utero [1]. According to different authors, infants learn their mother tongue quickly and without effort from six months to three years old, when they are capable of composing full sentences. Nonetheless, this is a complex process that need further analysis, since it has been shown that maturational and experiential factors play a key role in the development of language and human speech [2,3]. The development of language depends on brain maturation, which is modulated, among other environmental factors, by differences in infant diet. In this regard, early dietary factors can influence neurodevelopmental processes [4].

Long-chain polyunsaturated fatty acids (LC-PUFAs) intake during infancy is associated with optimal brain development. There is evidence that docosahexaenoic acid (DHA) intake and status are related to infant cognitive performance, including verbal learning ability, language, reading, spelling, nonverbal intelligence, and memory [5]. Infants fed formulas supplemented with DHA and arachidonic acid (ARA) performed better in executive function, vocabulary, and intelligence at three to five years of age compared to those who were fed with the formula with no DHA and ARA [6]. Additionally, bovine milk fat globule membrane (MFGM) has been added recently to infant formulas. MFGM is rich in sialic acid as part of gangliosides and glycosylated proteins, which have also been shown to be important for optimal brain development in different studies [7,8]. Furthermore, infant formulas are currently supplemented with probiotics and prebiotics, which are known for their role in the establishment, composition, and metabolic activity of gut microbiota [9]. Moreover, a relationship between gut microbiota and the brain has been proposed, which is known as gut microbiota-brain axis [10]. The extent of how this circuit modulates neurodevelopment and how different types of feeding exert an influence is still under investigation.

We hypothesized that supplementation during the first months of life with an infant formula enriched with functional nutrients, including LC-PUFAs (DHA and ARA), MFGM, and synbiotics, among others, might have long-term beneficial effects on language development in four-year-old children.

## 2. Materials and Methods

### 2.1. Study Design and Subjects

The current analysis included 122 Spanish children born at term from COGNIS Project (*A Neurocognitive and Immunological Study of a New Formula for Healthy Infants*) (registered at www.ClinicalTrials.gov as NCT02094547). Detailed information on this project, including the study design, subject recruitment, and population characteristics, were described elsewhere [11,12]. Briefly, COGNIS is a prospective, randomized, and double-blind study consisting in a nutritional intervention designed to evaluate the effects of a new infant formula on psychomotor, cognitive, socioemotional, and behavioral development. A total of 220 participants met the inclusion criteria. Of them, 170 were randomized into two groups to receive infant formula during their first 18 months of life: 85 infants were fed with a standard infant formula (SF) and 85 infants received an experimental infant formula (EF) enriched with in bioactive nutrients (MFGM, Fructooligosaccharides (FOS): Inulin proportion 1:1; *Bifidobacterium longum subsp. infantis* CECT7210 *(Bifidobacterium infantis* IM1) and *Lactobacillus rhamnosus* LCS-742), LC-PUFAs, gangliosides, nucleotides and sialic acid). Furthermore, 50 exclusively breastfed infants (BF) were included as a reference group.

A detailed participant flowchart from the baseline visit to four years old is shown in Figure 1. Excluding dropouts, 122 infants attended visit at four years old for the analysis of language development using the Oral Language Task of Navarra-Revised (PLON-R) test (SF (*n* = 46), EF (*n* = 43) and BF (*n* = 33)).

### 2.2. Ethics, Consent, and Permissions

This study was performed according to the Declaration of Helsinki II Principles [13], and all procedures were approved by the University of Granada Research Ethical Committee. The Bioethical Committees for Clinical Research of the Clinical University Hospital San Cecilio and the Mother-Infant University Hospital of Granada (Spain) also approved the project and protocols. All families were informed about all procedures and signed an informed consent before involving each child in the study.

### 2.3. Assessments of Children Language: Oral Language Task of Navarra-Revised (PLON-R)

PLON-R is a standardized test that allows an early detection or screening of the oral language development in kindergarten children. With a focus on the language dimensions (form, content, and use), with specific activities for each dimension, the scores of each one of the dimensions are transformed into typical punctuations organized in three categories by age: “Delay” (form ≤ 25; content ≤ 22; use ≤ 28; total ≤ 27), “need to improve” (form = 36; content: 33–47; use = 39; total: 39–45), and “normal” (form ≥ 50; content ≥ 67; use ≥ 59; total ≥ 54). Higher scores are related with better language development. Moreover, this test allowed us to obtain a total punctuation regarding language development [14].

### 2.4. Statistical Analysis

All statistical analyses were performed using the IBM^®^ SPSS Statistics^®^ program, version 22.0 (SPSS Inc. Chicago, IL, USA). Normally distributed variables were presented as mean and standard deviation (SD), and non-normal variables as median and interquartile range (IQR). Categorical variables were showed as frequencies and percentages. Differences in language test scores among BF, SF, and EF groups were contrasted using analysis of variance (ANOVA) or Welch’s t-test, Kruskal–Wallis rank-sum test for non-normal continuous variables, and chi-square or Fisher test for categorical variables. In the event of significant group differences, Bonferroni-corrected *post hoc* comparisons were used to identify significant pair-wise group differences (corrected *p*-values < 0.05). In addition, variables showing significant group differences, such as maternal age and educational level, paternal age and educational level, sex of the child, and socioeconomic status, were included into analysis of covariance (ANCOVA).

Additionally, the Wald test for logistic regression was performed to calculate the odds ratios (ORs) and 95% confidence intervals (CI) of having normal or delayed values after adjustment for potential confounders. *p* values < 0.05 were considered statistically significant.

## 3. Results

### 3.1. Characteristics of the COGNIS Study Participants at Four Years Old

Background and baseline characteristics of 122 parents and children participating in the current study at four years old in the follow-up visit are shown in Table 1. The analysis revealed that mothers and fathers of BF children were older (*p* = 0.019 and *p* = 0.027, respectively) than parents from the SF group. Mothers from the BF group showed a higher educational level (*p* = 0.001) than those from EF and SF groups. Maternal IQ was higher in the BF group than mothers from the EF group (*p* = 0.023). Fathers from the BF group showed also a higher educational level in comparison with fathers from the SF group (*p* = 0.040). Furthermore, the BF group presented a higher socioeconomic status than the SF and EF groups (*p* < 0.001). Finally, there were significantly more girls than boys in the BF group compared to the SF group (*p* = 0.034). It is important to note that these differences did not exist between the SF and EF groups. Interestingly, no differences were observed in children IQ, bilingual children, and need for speech therapy at four years old.

### 3.2. Effects of Infant Formulas on Language Development in COGNIS-Children at Four Years Old

Afterward, the effects of both infant formulas and breastfeeding on language development were analyzed, comparing the PLON-R scores between the study groups (Table 2). At four years old, SF children seemed to present lower scores in language content (*p* = 0.026) and total score of PLON-R test (*p* = 0.029) compared to BF children in an unadjusted model. However, after adjustment for selected confounding variables (maternal age, educational level, and IQ, paternal age and educational level, sex of the child, and socioeconomic status), these differences disappeared. Interestingly, children who received EF seemed to show higher scores in the use of language (*p* = 0.033) and oral spontaneous expression (*p* = 0.024) than children who received SF. After adjustment for the above selected confounding variables, the use of language (*p* = 0.035) and oral spontaneous expression (*p* = 0.014) remained significant.

We next evaluated the association between the type of feeding and PLON-R scores categorized in each scale into normal or need to improve/delayed outcomes according to the standards test (Table 3). The analysis suggests showed that children fed with SF were more frequently categorized into delayed/need to improve/in use of language (*p* = 0.020) in comparison with children fed with EF. Moreover, the SF group seemed to present more frequently delayed/need to improve in PLON-R total score (*p* = 0.027) than children who were breastfed.

Finally, in order to evaluate the influence of other confounder variables, the Wald test for logistic regression was performed (Table 4). Maternal age, educational level, and IQ, paternal age and educational level, socioeconomic status, sex of the child, and the three study groups were included in this model. The type of feeding during the first months of life (COGNIS groups) might have an effect on use of language at four years old. In fact, it seems that SF-fed children presented an increased risk of suffering problems in the use of language (OR: 8.500 (95% CI: 1.504–48.049); *p* = 0.015) compared to BF children. Furthermore, a marginally significant *p*-value was found with respect to socioeconomic status. In this matter, children who had lower socioeconomic status might present an increased risk of suffering problems in the content of language (OR: 3.583 (95% CI: 0.977–13.148); *p* = 0.054) and PLON-R total scores (OR: 4.821 (95% CI: 0.966–24.063); *p* = 0.055). No other effects of other confounder variables on language development at four years old were found in the analysis performed.

## 4. Discussion

The present study suggests the long-term beneficial effects of a new infant formula enriched with functional nutrients on children’s language development at four years of age. In fact, children fed with the EF seemed to show better scores in use of language and oral spontaneous expression compared to those fed with the SF. Additionally, the SF group might be associated with worse scores in PLON-R total score compared to the BF group, with scores of the EF children similar to those who were breastfed. Finally, the use of language in children up to four years could be influenced by type of feeding during the first eighteen months and, to a lesser extent, by socioeconomic status. Thus, our results suggest that the addition of bioactive nutrients to the new infant formula improves children’s language development, achieving similar scores to breastfed children.

Nutrition during the first months of life helps to build brain cells, establishing connections between them and creating brain architecture for optimal neurodevelopment [15] and language acquisition [16]. One of the most accepted mechanism refers to differences in fatty acid profile of breast milk and infant formula, which might have a different effect on brain structure and function [17]. Other potential mechanisms influencing neurodevelopment could be the impact of breastfeeding on the own mother, and the special mother–infant interaction, which could shape the learning and language skills of their offspring [18]. Furthermore, neurodevelopment is the result of the dynamic interrelation between environment and genetic background. Exposure to adverse experiences can negatively affect gene expression, as well as brain development and its cognitive and behavioral functions. If this occurs during critical developmental periods, the effects could be permanent [19].

Breast milk is considered the gold standard for infant nutrition and appears to be beneficial for an adequate neurological development [20]. This effect could be caused by specific bioactive nutrients and other components present in human milk, which promote a synergistic effect and are absent or present in lower amounts in infant formulas [21,22]. Recently, it has been demonstrated that breastfed children have better language development, stimulus processing, and increased IQ during the first year of life compared to formula-fed infants [23].

LC-PUFAs, especially DHA and ARA, are found in high proportions in the structural lipids of cell membranes, particularly those of the central nervous system [24]. Interestingly, it has been reported that the LC-PUFAs content of brain cell membranes could influence gene expression within those cells [25]. DHA is considered essential for the cortical circuit maturation in the developing brain, and ARA is also critical for childhood growth, brain development and health [26]. In this matter, infants who received a supplement containing ARA (200 mg/day) and DHA (200 mg/day) from 12–24 months of age showed better cognition and language skills compared to those without supplementation [27]. An interesting issue is that the EF tested in the COGNIS study was supplemented with ARA (0.45%) and DHA (0.32%), while the SF only contained linoleic and α-linolenic acids. EF seemed to be associated with better language development at four years of age. Furthermore, higher DHA levels have been associated with larger cortical grey matter and cerebellar volumes [28], which are involved in different language tasks [29]. Several randomized clinical trials (RCTs) have also shown that infants supplemented with DHA show better language and communication skills [30,31]. On the other hand, research efforts have focused on infant formula supplementation with MFGM, which is a biologically active component derived of milk proteins present in human milk [32], including a lipid fraction consisting of LC-PUFAs (also ARA and DHA) [33]. In this regard, addition of MFGM to an infant formula has been found to improve immune system and neurodevelopment during the first months of life [8,34].

Results of the current study support the importance of the type of infant nutrition on brain development, particularly on language skills. So far, to our knowledge, the majority of studies have focused on determining the effect of a single component added to an infant formula on language development and cognitive function. In this regard, our study is pioneering because it evaluates the whole effects of several functional nutrients that help to narrow the gap between the composition of the EF and breast milk. Our current results suggest that language development in EF children is similar to BF children. However, it is difficult to ascertain whether this effect is due to a single nutrient or to the synergistic effect of all of them. Although a possible positive and direct effect of LC-PUFAs has been suggested, we do not know the potential role of other bioactive compounds, including MFGM, gangliosides, nucleotides, or sialic acid, on language development, as well as the potential interactions between them.

Furthermore, it is important to note the long-term nutritional intervention performed in the COGNIS study were carried out in the infants during their first 18 months of life, which it is established as a fundamental period of brain maturation and growth [11,35]. It is known that language development is related to brain maturation during the first two years of life, together with the maturation of the brain circuits that connect Broca’s and Wernicke’s areas (fronto-temporo-parietal language system) [36]. Previous studies have shown a greater volume of white and gray matter in the left and right parietal and left temporal lobes, as well as more activation in the right frontal and left temporal lobe in breastfed children between five and seven years old, evidencing mechanisms linked to breastfeeding and neurocognitive outcomes [20,37,38]. Therefore, the functionality and connectivity in the fronto-temporo-parietal language system could be the key for language acquisition and processing during early life [15,16,17].

It is important to note that logistic regression models seemed to show a trend of poorer language skills in children with low socioeconomic status, regardless the type of feeding received. There are different factors, such as genetic background, sex, intrauterine growth restriction, nutrition, maternal education, and socioeconomic status, among others, that have an important role during brain development [39]. Others researchers have reported that children born in families with lower socioeconomic status showed slower language development [40,41]. Furthermore, the results of the current study are in agreement with other study reporting that the beneficial effects of breastfeeding on language skills in children at five years old were reduced when social and parental factors were taken into account [42]. However, further studies should be performed in order to confirm this outcome.

The main strengths of the COGNIS study are the design, considering that is a prospective RCT with a long-term intervention and monitoring period. Moreover, several confounding factors, which may influence neurodevelopment, were taken into account in the statistical models, to provide accurate, consistent, and reliable results and conclusions. Our results could provide evidence of the long-term effects of early nutrition on language development in healthy children, while the vast majority of previous studies have focused on language pathologies. We used the PLON-R test, a highly reliable and valid instrument for assessing language development in Spanish children [14]. However, even if language tests are validated and adapted to children at four years of age, it is quite difficult to obtain significant differences in healthy children at this age, which could be a limitation of the study. Nevertheless, the statistical power at this time point on the COGNIS study reached 80%, allowing the authors to detect a minimum difference of 0.7 SD, which is enough to detect relevant differences in language development between the study groups.

## 5. Conclusions

The findings from this study suggest the long-term beneficial effects of a new infant formula containing LC-PUFAs, MFGM, synbiotics, and other functional nutrients on language development in healthy children at four years old. Thus, this new infant formula might promote optimal brain development in a similar way to breast milk. Furthermore, socioeconomic factors seem to be involved in language development during childhood.

## Figures and Tables

**Figure 1 nutrients-12-00535-f001:**
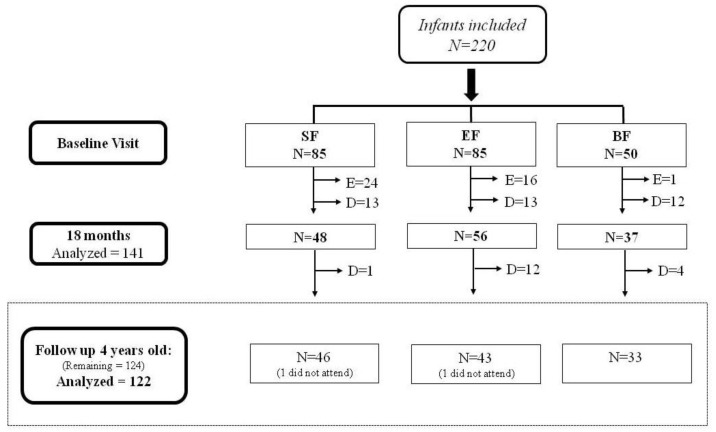
Participant flowchart from baseline visit to four years old. SF: Standard infant formula; EF: Experimental infant formula; BF: Breastfed infants; D = Dropouts; E = Exclusions. Up to 18 months of life, a total of 40 infants were excluded in the SF and EF groups as follows: 24 were excluded in the SF group (1 infant due to perinatal hypoxia, 1 infant had growth deficiency, 15 infants did not take the infant formula, 2 had colic of the infant, 3 were excluded due to lactose intolerance, 1 infant due to digestive surgical intervention, and 1 infant suffered hydrocephalus); 16 infants were excluded in the EF group (2 infants presented growth deficiency, 2 infants lactose intolerance, 11 infants did not take the infant formula, and 1 was excluded due to epileptic seizure). Furthermore, one infant of the BF group was excluded because he/she was not breastfed. During the follow-up visit at four years old, dropouts were considered those who did not continue to participate in the study; *n* = sample size.

**Table 1 nutrients-12-00535-t001:** Baseline characteristics of children and parents participating in the COGNIS project.

		Follow up Four Years Old			
Parents characteristics		SF (*n* = 46)	EF (*n* = 43)	BF (*n* = 33)	*p* ^1^
Maternal age (years)		31.00 (24.00–35.00) ^a^	31.00 (28.00–34.00) ^a,b^	34.00 (31.00–38.00) ^b^	**0.019**
Maternal pBMI (kg/m^2^)		24.17 (21.05–26.30)	25.15 (22.21–28.48)	24.46 (23.05–25.95)	0.478
Maternal educational level	Primary	6 (13.04%)	9 (20.93%)	1 (3.03%)	**0.001**
	Secondary	12 (26.09%) ^a,b^	13 (30.23%) ^b^	2 (6.06%) ^a^	
	VT	13 (28.26%)	15 (34.88%)	9 (27.27%)	
	University	15 (32.61%) ^a^	6 (13.95%) ^a^	21 (63.64%) ^b^	
Maternal IQ (points)		104.00 (95.00–112.50) ^a,b^	100.00 (89.00–108.00) ^a^	111.00 (96.00–117.00) ^b^	**0.023**
Smoking during pregnancy	No	36 (80.00%)	36 (83.72%)	31 (93.94%)	0.220
	Yes	9 (20.00%)	7 (16.28%)	2 (6.06%)	
GWG (kg)		7.00 (3.50–10.00)	7.00 (4.00–10.00)	6.25 (4.50–8.50)	0.758
Type of delivery	Vaginal	34 (73.91%)	31 (72.09%)	25 (75.76%)	0.937
	Cesarean	12 (26.09%)	12 (27.91%)	8 (24.24%)	
Postpartum Depression	No	35 (76.09%)	34 (80.95%)	28 (84.85%)	0.621
	Yes	11 (23.91%)	8 (19.05%)	5 (15.15%)	
Paternal age (years)		32.50 ± 6.69 ^a^	33.54 ± 5.56 ^a,b^	36.24 ± 4.38 ^b^	**0.027**
Paternal educational level	Primary	11 (25.00%)	16 (39.02%)	6 (18.18%)	**0.040**
	Secondary	16 (36.36%)^a^	9 (21.95%) ^a,b^	4 (12.12%) ^b^	
	VT	8 (18.18%)	9 (21.95%)	10 (30.30%)	
	University	9 (20.45%)	7 (17.07%)	13 (39.39%)	
Paternal IQ (points)		106.86 ± 12.38	104.29 ± 15.81	106.90 ± 12.93	0.659
Socioeconomic status	Low	9 (19.57%) ^a,b^	10 (23.81%) ^b^	1 (3.03%) ^a^	**<0.001**
	Middle-Low	22 (47.83%) ^a^	25 (59.52%) ^a^	6 (18.18%) ^b^	
	Middle-High	12 (26.09%) ^a,b^	4 (9.52%) ^b^	17 (51.52%) ^a^	
	High	3 (6.52%) ^a^	3 (7.14%) ^a,b^	9 (27.27%) ^b^	
Place of residence	Urban	20 (44.44%)	11 (26.19%)	8 (24.24%)	0.095
	Rural	25 (55.56%)	31 (73.81%)	25 (75.76%)	
Siblings	0	15 (33.33%)	18 (42.86%)	8 (24.24%)	0.238
	≥1	30 ± 66.67	24 ± 57.14	25 ± 75.76	
Gestational age (weeks)		40.00 (39.00–41.00)	40.00 (39.00–41.00)	40.00 (39.00–41.00)	0.697
**Newborn and child characteristics**					
Birth weight (g)		3402.89 ± 402.49	3418.84 ± 482.08	3374.24 ± 392.58	0.904
Birth length (cm)		51.00 (50.00–52.00)	51.00 (50.00–52.00)	51.00 (50.00–52.00)	0.678
Birth head circumference (cm)		35.00 (34.00–36.00)	34.00 (34.00–35.00)	35.00 (34.00–35.00)	0.155
Children sex	Boy	31 (67.39%) ^a^	27 (62.79%) ^a,b^	13 (39.39%) ^b^	**0.034**
	Girl	15 (32.61%) ^a^	16 (37.21%) ^a,b^	20 (60.61%) ^b^	
Children IQ (points)		111.50 (97.00–118.00)	113.00 (103.00–120.00)	115.00 (105.00–121.00)	0.551
Children BMI/Age	Severe thinness	0 (0.00%)	0 (0.00%)	0 (0.00%)	0.388
	Thinness	0 (0.00%)	0 (0.00%)	0 (0.00%)	
	Adequate weight	33 (71.74%)	25 (58.14%)	26 (78.79%)	
	Risk of overweight	9 (19.57%)	13 (30.23%)	4 (12.12%)	
	Overweight	4 (8.70%)	4 (9.30%)	3 (9.09%)	
	Obesity	0 (0.00%)	1 (2.33%)	0 (0.00%)	
Bilingual		2 (4.30%)	3 (7.00%)	2 (6.10%)	0.890
Speech therapy		5 (10.90%)	1 (2.30%)	1 (3.00%)	0.264

Data are presented as *n*(%) for categorical data, mean ± SDs for parametrically distributed data, and median (IQR) for nonparametrically distributed data. ^1^
*p*-values for overall differences between COGNIS study groups. *p*-values were obtained from ANOVA for normally distributed variables, Kruskal–Wallis rank-sum test for non-normal continuous variables, and chi-square test for categorical variables. Values not sharing the same suffix (ab) were significantly different in the Bonferroni post hoc test. *p*-values < 0.05 are highlighted in bold. BF: Breastfed infants; EF: Experimental infant formula; GWG: Gestational weight gain; IQ: Intelligence quotient; pBMI: Pre-conceptional body mass index; SF: Standard infant formula; VT: Vocational training.

**Table 2 nutrients-12-00535-t002:** Effects of infant formulas on language development at four years old.

PLON-R Scores	SF (*n* = 46)	EF (*n* = 43)	BF (*n* = 33)	*P_unadj_*	F (df)	*P_adj_*	η_p_^2^
Form ^1^	54.85 ± 18.18	56.49 ± 16.92	59.39 ± 15.28	0.503	0.387 (2,102)	0.680	0.008
Phonology	0.80 ± 0.54	0.81 ± 0.39	0.91 ± 0.29	0.376	2.006 (2,102)	0.140	0.038
Morphology-Syntax	3.28 ± 1.05	3.40 ± 0.85	3.48 ± 0.87	0.630	1.285 (2,102)	0.281	0.025
Sentences repeat	1.48 ± 0.75	1.44 ± 0.70	1.58 ± 0.71	0.717	0.850 (2,102)	0.431	0.016
Oral Spontaneous Expression	1.8 ± 0.40	1.95 ± 0.30	1.91 ± 0.29	0.512	1.493 (2,102)	0.230	0.028
Content ^1^	41.50 ± 16.64 ^a^	42.84 ± 18.08 ^a,b^	51.42 ± 15.05 ^b^	**0.026**	0.826 (2,102)	0.441	0.016
Lexicon	1.20 ± 0.58	1.35 ± 0.65	1.52 ± 0.51	0.062	1.091 (2,102)	0.340	0.021
Comprehension	0.89 ± 0.31	0.91 ± 0.29	1.00 ± 0.00	0.161	1.080 (2,102)	0.343	0.021
Expression	0.30 ± 0.47	0.44 ± 0.50	0.52 ± 0.51	0.149	0.707 (2,102)	0.496	0.014
Colors identification	0.98 ± 0.15	1.00 ± 0.00	0.97 ± 0.17	0.557	1.442 (2,102)	0.241	0.028
Spatial relations	0.85 ± 0.36	0.77 ± 0.43	0.94 ± 0.24	0.078	0.266 (2,102)	0.767	0.005
Opposites	0.61 ± 0.49	0.65 ± 0.48	0.82 ± 0.39	0.090	0.539 (2,102)	0.585	0.010
Basic needs	0.78 ± 0.42	0.72 ± 0.45	0.88 ± 0.33	0.207	0.048 (2,102)	0.953	0.001
Use ^1^	46.00 ± 11.35 ^a^	51.86 ± 11.15 ^b^	50.79 ± 10.51 ^a,b^	**0.033**	3.461 (2,102)	**0.035**	0.064
Oral Spontaneous Expression (picture)	1.89 ± 0.38	1.93 ± 0.34	1.91 ± 0.29	0.866	2.376 (2,102)	0.098	0.045
Oral Spontaneous Expression (puzzle)	0.4 ± 0.50 ^a^	0.70 ± 0.46 ^b^	0.67 ± 0.48 ^a,b^	**0.024**	4.472 (2,102)	**0.014**	0.081
PLON-R Total Score ^1^	47.07 ± 16.90 ^a^	51.23 ± 18.12 ^a,b^	57.48 ± 15.18 ^b^	**0.029**	1.512 (2,102)	0.225	0.029

Data are presented as means ± SD of direct scores. ^1^ Data are presented as means ± SD of typical scores. *P_unad_*_j_ is ANOVA. Values not sharing the same suffix (ab) were significantly different in a Bonferroni post hoc test. *P_adj_* is ANCOVA for the group differences using the univariate general linear model, including confounder factors: Maternal age, educational level, and IQ, paternal age and educational level, sex of the child, and socioeconomic status. F-values (F) and effect sizes (η_p_^2^) were calculated by ANCOVA. *p*-values < 0.05 are highlighted in bold. BF: Breastfed infants; df: Degrees of freedom; EF: Experimental infant formula; PLON-R: Oral Language Task of Navarra-Revised; SF: Standard infant formula.

**Table 3 nutrients-12-00535-t003:** Association between infant formulas and PLON-R clinical clusters in children at four years old.

PLON-R Scales		SF (*n* = 46)	EF (*n* = 43)	BF (*n* = 33)	Chi-Square	*P* ^1^
Form	Delayed/Need to improve	13 (28.26)	9 (20.93)	4 (12.12)	2.991	0.224
	Normal	33 (71.74)	34 (79.07)	29 (87.88)		
Content	Delayed/Need to improve	37 (80.43)	32 (74.42)	20 (60.61)	2.991	0.142
	Normal	9 (19.57)	11 (25.58)	13 (39.39)		
Use	Delayed/Need to improve	28 (60.87) ^a^	14 (32.56) ^b^	13 (39.39) ^a,b^	7.786	**0.020**
	Normal	18 (39.13) ^a^	29 (67.44) ^b^	20 (60.61) ^a,b^		
PLON-R Total Score	Delayed/Need to improve	25 (54.35) ^a^	19 (44.19) ^a,b^	8 (24.24) ^b^	7.188	**0.027**
	Normal	21 (45.65) ^a^	24 (55.81) ^a,b^	25 (75.76) ^b^		

Data are presented as *n*(%). ^1^
*p*-values for overall differences between COGNIS study groups. *p*-values were obtained from the chi-square test for categorical variables. Values not sharing the same suffix (ab) were significantly different in the Bonferroni *post hoc* test. *p*-values < 0.05 are highlighted in bold. BF: Breastfed infants; EF: Experimental infant formula; PLON-R: Oral Language Task of Navarra-Revised; SF: Standard infant formula. Categories according to standards PLON-R: Normal (form: ≥ 50; content: ≥ 67; use: ≥ 59; total: ≥ 54) and delayed/need to improve (form: ≤ 36; content: ≤ 47; use: ≤ 39; total: ≤ 45).

**Table 4 nutrients-12-00535-t004:** Effects of COGNIS groups and socioeconomic status on language outcomes at four years old.

PLON-R Scales	SF Group * OR (95% CI)	*p*	Socioeconomic Status ^†^ OR (95% CI)	*p*
Form	-	-	-	-
Content	-	-	3.583 (0.977–13.148)	0.054
Use	8.500 (1.504–48.049)	**0.015**	-	-
Total Score	-	-	4.821 (0.966–24.063)	0.055

Model was a logistic regression (Wald method). Dichotomized scales for binary logistic regression: normal and delayed/need to improve. *p*-values < 0.05 are highlighted in bold. * Breastfeeding group as reference. The SF group was not included in the model of form, content, and total score. ^†^ High socioeconomic status as reference. Socioeconomic status was not included in the model of form and use. CI: Confidence interval; OR: Odds ratio; PLON-R: Oral Language Task of Navarra-Revised; SF: Standard infant formula.

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
