# Peer review of "Influence of a Functional Nutrients-Enriched Infant Formula on Language Development in Healthy Children at Four Years Old"

_nutrients, 2020, doi:10.3390/nu12020535_

Round 1

Reviewer 1 Report

Nieto-Ruiz et al. have conducted a study into the relationship between dietary intake of neonates and CNS development as measured by cognition in childhood. 

This type of study is difficult to run and is lacking in the literature.  The numbers of participants place this study in the category of pilot study rather than a typical RCT, but this only requires clear explanation by the authors and is not a flaw in itself. 

I am concerned about the conflict of interest of two of the research team.  I am pleased to see that this is clearly and honestly stated.  I recommend that the roles of these authors in the research is clarified in greater detail, e.g. through an Author Contributions statement.

There is at least one paper that has come out in the last year that begins to explore the relationship between what is in the mother’s system (and thus what she eats) and thus which fats/lipids there are in her milk and then the infant’s blood.  However, the diet of mothers who breastfed does not appear to be recorded or discussed.  This area is worthy of some discussion as it implies there is some importance of maternal intake in child development post partum

My other concern regards the significance values.  P-values below 0.05 are highlighted.  However, there does not appear to have been a correction for multiple testing.  The number of tests presented in Table 2 for example, strongly suggests that 0.05 is not an appropriate threshold in order to conclude that there is an association.  However, I think it is reasonable to say that there may be a relationship that would pass an FDR-corrected threshold with sufficient statistical power, but that further work is required to demonstrate this (e.g.  a full RCT with 5k+ participants).  I therefore recommend that the wording throughout the manuscript is modified to reflect a more circumspect conclusion.  It is not possible from these data to make a decision about which method of feeding is 'best'.

Finally, I think it is important to comment on the effect size.  For example, the values for spontaneous expression are low but the percentage difference may be high.  I would like to see a clearer explanation of what this means.

Reviewer 2 Report

Review: Influence of a functional nutrients-enriched infant formula

This paper compared language development at the age of 4 for those infants who were either breastfed, fed by standard formula milk, or with enriched formula milk in the first stages of infancy. It was a randomized double-blind study. Results show that infants in the enriched group develop better language skills than infants fed with the standard formula milk. I liked the paper, and most of the results and discussion were sound. In general, the paper was well –written (there are some grammatical errors that a native reader should easily spot). To improve the paper, I summarize my comments below, but most of them I’d consider minor (e.g., requests for clarifications) and should be relatively easy to adjust.

Introduction

Your references to how quickly infants acquire their NATIVE language make little sense. I was expecting references to works by Patricia Kuhl (Kuhl, P. K. (2004). Early language acquisition: cracking the speech code. Nature reviews neuroscience, 5(11), 831-843.), or Janet Werker (Werker, J. F., & Hensch, T. K. (2015). Critical periods in speech perception: new directions. Annual review of psychology, 66, 173-196.). I would also stress that it is not just language, but learning a native language.

Methods:

You write that feeding groups maintained their diet for 18 months (cf. abstract). How did you check that this was the case? Surely there must be some women in the breast-feeding group who stopped within the first year? Is there any information how long infants continued with their diet? Do the groups also differ in how much solid food they get in the 18 months (when they slowly and increasingly switch to food instead of just milk)? In other words, besides a categorical variable, is there any information about the quantity and duration of their diet?

A possibly related question to the point above is why there is such a large dropout in the EF-group at 18 months. In the figure heading you list reasons for exclusions across groups; I’d rather see them separate for each group.

When you describe the scoring of the oral task in 2.3, it would help to have a line saying the higher the scores, the better their language development. Also the cut-off for form/use-aspect is one number (e.g. 36), should it not be a range?

When you report p-values, it is now standard to report F-values or other test statistiscs as well, plus effect sizes.

@Table 1:

- sometimes brackets refer to range, other times to percentages. This is unclear. When talking about proportions, just add % within the brackets (as in with children’s sex).

- we need more information about the children’s IQ task. When was it administered? What kind of task? Also those infants who were non-Spanish native – what does this mean? Are they bilingual, and if so simultaneous/sequentially?

Results

Table 3: unclear how p-values are obtained. Figure headings below suggest Chi-square tests, but methods says Wald tests, adjusted for confounds. Could you be more explicit here? Did yu also control for confounding effects in the post-hoc Bonferroni tests?

Discussion:

First paragraph: I would tone down the conclusion a bit, since most results do not survive after controllling for confounders.  

2nd paragraph: You do not find increased IQ for breastfed children in the current study – I wonder why.

Line 256: first 18 months crucial for brain development – needs references. Like work by Mark Johnson.

Line 278: similar, a reference is required for the assumption that PLON-R is highly reliable. Would you expect similar findings for a task measuring language perception skills?

Final paragraph (282-285): I do not agree with lack of differences in language development in individual differences – see work by Evan Kidd for instance  - they exist, they are meaningful. This is why we need effect sizes to interpret the contribution of diet to language development. You either need to drop this reasoning, or expand more. Moreover, the 80% power – where does this come from? Could you expand?

Conclusions:

Can you say that SES is a separate factor besides nutrition, or is  nutrition a mediating factor from SES to language development? I would welcome a more thorough discussion on this.

--- the end.

.
